# An Effective Model of Confidentiality Management of Digital Archives in a Cloud Environment

**Jian Xie [1], Shaolong Xuan [1,*], Weijun You [2,*], Zongda Wu [1,*] and Huiling Chen [3]**

[1] Deparment of Computer Science and Engineering, Shaoxing University, Shaoxing 312000, China
[2] Department of Management, Office of Natural Science Foundation of Zhejiang Province, Hangzhou 310006, China
[3] College of Computer Science and Artificial Intelligence, Wenzhou University, Wenzhou 325035, China
* Correspondence: zjsxxsl@163.com (S.X.); uweijun@zjinfo.gov.cn (W.Y.); zongda1983@163.com (Z.W.)

**Abstract:** Aiming at the problem of confidentiality management of digital archives on the cloud, this paper presents an effective solution. The basic idea is to deploy a local server between the cloud and each client of an archive system to run a confidentiality management model of digital archives on the cloud, which includes an archive release model, and an archive search model. (1) The archive release model is used to strictly encrypt each archive file and archive data released by an administrator and generate feature data for the archive data, and then submit them to the cloud for storage to ensure the security of archive-sensitive data. (2) The archive search model is used to transform each query operation defined on the archive data submitted by a searcher, so that it can be correctly executed on feature data on the cloud, to ensure the accuracy and efficiency of archive search. Finally, both theoretical analysis and experimental evaluation demonstrate the good performance of the proposed solution. The result shows that compared with others, our solution has better overall performance in terms of confidentiality, accuracy, efficiency and availability, which can improve the security of archive-sensitive data on the untrusted cloud without compromising the performance of an existing archive management system.

**Keywords:** cloud; digital archives; confidentiality management; information system

## 1. Introduction

In cloud computing, pay-per-use enables an organization to obtain the required sources from the shared pool of configurable computing resources anytime, anywhere and on-demand [1–3], therefore, greatly reducing the organization's expenditure on business operations and archive management and then improving the service efficiency of the organization. To this end, governments and enterprises in various countries have promoted the cloud-first strategy [4–6], i.e., the cloud computing model is given priority in the process of institutional informatization, such that the proportion of archival documents formed and managed on the cloud is becoming higher and higher. Archive management on the cloud has become the general trend [7–9]. However, although storing digital archives on the cloud can reduce the management cost and improve management efficiency, it also results in some negative effects, the most prominent of which is the security of archives on the cloud [10–12]. In a cloud computing environment, the archives of an organization are not stored on a trusted local server but are stored and managed by the cloud server, resulting in each archive and its owner being separated from each other, i.e., making each archive in an uncontrollable area, and in turn posing a serious threat to the security of archival materials [13–15]. Such security threat mainly includes two aspects: (1) external threat, i.e., hackers' attack on the cloud service provider (which has been verified by endless hacking incidents) [16]; and (2) internal threat, i.e., inside jobs from workers of the cloud service provider (driven by interests, it is possible for management workers to maliciously stealing sensitive archival information) [17]. In a word, the security issue of archives on the

cloud (i.e., how to ensure the security of sensitive archival data on the untrusted cloud) has become one of the main obstacles to restricting the management of archives on the cloud, which has attracted more and more attention.

### 1.1. Related Works and Limitations

Aiming at the problem of the security of archives on the cloud, scholars from the field of social sciences conducted more research from the perspective of laws and regulations and believe that the solution of the problem requires governments to formulate relevant laws and regulations for guidance [18,19]. Most of the countries in the world have successively formulated relevant standards and specifications, such as the Guideline for Document Management in Cloud Computing Environment in the United States, Advice on Risk Management of Cloud Computing File Storage in Australia, and Guidelines for Cloud Storage and Digital Permanent Preservation in the United Kingdom. In recent years, China has also intensively promulgated three relevant laws and regulations, i.e., Cybersecurity Law, Data Security Law, and Personal Information Protection Law, which play an important role in ensuring the security of archives on the cloud [20,21]. However, endless incidents of privacy breaches show that confidentiality management of archives on the cloud requires not only laws and regulations, but also the support of technical methods [22–25].

In order to ensure the security of archive data, a digital archive management system uses a variety of technical methods and strategies, such as identity authentication, access control and data encryption. Below, we briefly introduce the technical features of these methods and analyze their application limitations in the confidentiality management of archives on the cloud. (1) Identity authentication is the process of user identity confirmation, to prevent illegal users from accessing system resources illegally [11,26]. Specifically, it can be divided into two categories, i.e., single-factor authentication [27–29] (such as username and password authentication, smart card authentication, dynamic password authentication and biometric authentication) and two-factor authentication [30,31] (which combines two kinds of single-factor authentication to further strengthen the security of identity authentication). (2) Access control is to restrict access to unauthorized resources or the use of unauthorized functions, according to the specific identity of a user [32]. Specifically, it can be divided into discretionary access control (DAC) and mandatory access control (MAC) [33]. Identity authentication and access control have been widely used in operating systems, database systems, file management systems [34], etc. Although the two kinds of technical methods can prevent external users from illegally accessing the sensitive data in a digital archives system, to alleviate the archive security problem to a great extent, they cannot separate the support of the server side (they assume that the server side is trusted), i.e., they only target external illegal attackers of a digital archive system, and cannot prevent the internal staff of the untrusted server side (or the hackers who conquer the server) from accessing the archive-sensitive data [35–37]. However, the cloud is not trustworthy, and it is the main source causing the archive security problem. Therefore, the problem of confidentiality management of archives on the cloud cannot be solved by traditional access control and identity authentication. (3) Data encryption refers to strictly encrypting sensitive data before being stored in an untrusted server, so that even if the encrypted data is leaked, it is difficult to be understood, consequently, ensuring data security [38,39]. Therefore, it is an important means to solve the data security problem in a cloud environment [40–43]. However, there are a large number of query operations defined over archive-sensitive data in a digital archive system (such as querying archives by user names). Once the sensitive data stored on the cloud is strictly encrypted, the ciphertext data would lose many inherent characteristics of the corresponding plaintext data (such as orderliness, similarity and comparability), resulting in most of the original archive query operations in an archive system no longer being performed correctly on the ciphertext data, consequently, damaging the accuracy of archival search [44,45]. In order to solve the ciphertext search problem, we can first transmit all the ciphertext data on the cloud back to a local server, decrypt the ciphertext data, and then perform archive query operations on

the decrypted data. However, for such a method, since almost all the process of archive search is completed locally, it not only completely loses the cost-efficiency advantage of archive management on the cloud, but also seriously reduces the efficiency of archive search (it needs huge overhead for network transmission and decryption). Therefore, the problem of confidentiality management of digital archives on the cloud cannot be directly solved by a traditional data encryption method.

In addition, scholars from the field of library science also try to solve the problem of the security of archives on the cloud from the perspective of technical methods [46–48]. However, the methods proposed by them are usually developed based on some original technical methods from a digital archive management system (i.e., identity authentication, access control, data encryption, etc.), so it is difficult to meet the actual needs of confidentiality management of archives on the cloud. For the problem of cloud data security, scholars from the field of information sciences have also conducted in-depth and systematic research and proposed many effective technical methods [49–53]. However, these methods are not specifically proposed for digital archives systems, so they still cannot meet the practical application requirements of confidentiality management of archives on the cloud in terms of availability, effectiveness and security. To sum up, under the existing architecture of a digital archive cloud management platform, it remains to be further discussed and studied how to improve the security of archive-sensitive data on the untrusted cloud without compromising the availability of an archive system and the effectiveness of archive search.

### 1.2. Contributions

In this paper, we propose an effective solution for confidentiality management of archives on the cloud, which can improve the security of sensitive archive data on the cloud without affecting the efficiency of archive search. Its basic idea is to deploy a local server between the cloud and each client of an archive system to run a confidentiality management model of digital archives on the cloud (specifically, which includes an archive release model and an archive search model), which acts as a layer of middleware between the cloud and the client, to achieve transparency for users and the cloud, and then achieve effective integration with the existing archive management system. Specifically, the contributions of this paper mainly include the following three aspects. (1) Propose a confidentiality release model of archives on the cloud, which is responsible for strictly encrypting the archive files and archive data released by an administrator, generating archive feature data for the archive data, and then submitting them to the cloud for storage to ensure the security of archive data. (2) Propose a confidentiality search model of archives on the cloud, which is responsible for rewriting and transforming the query operations defined on archive data submitted by an inquirer, so that it can be correctly executed on feature data on the cloud (to filter out most of the non-target records) to ensure the accuracy and efficiency of archive search. (3) Both theoretical analysis and experimental evaluation demonstrate the overall performance of the proposed solution, i.e., it can satisfy the actual requirements in terms of data security, query accuracy, and query efficiency. This paper gives a valuable study attempt on confidentiality management of archives on the cloud, which has positive significance for promoting the application and development of cloud computing technology in digital archives management.

## 2. Problem Statement

### 2.1. System Framework

In Figure 1, we show the basic framework of a confidentiality management model of digital archives on the cloud adopted in this paper. It can be seen that it mainly includes the following four roles, i.e., archive administrators and their management interfaces (trusted), archive inquirers and their query interfaces (trusted), a local server (trusted) and the cloud server (untrusted). The functions of the four types of roles are briefly described below.

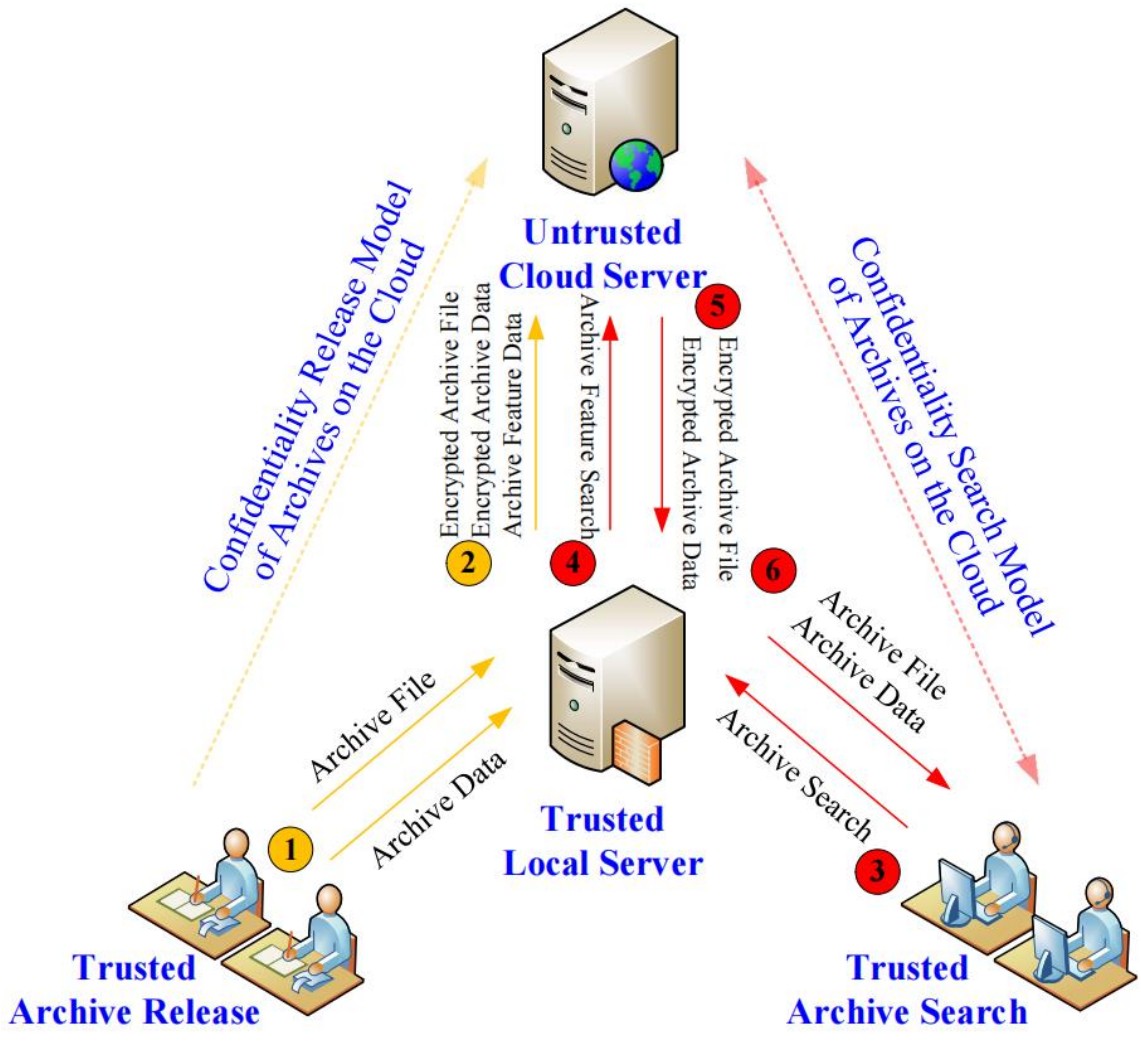

**Figure 1.** A framework of a confidentiality management model of archives on the cloud.

1. Archive administrator (also known as archive entry clerk): through a trusted archive management interface, who submits digital archive files (electronic scanning pictures) and their corresponding archive data (usually in the form of tables, which are used to record archive description data to facilitate archive search).
2. Archive inquirer: through a trusted archive search interface, who performs archive search operations (i.e., perform related archive query operations defined on archive description data) to obtain target archive files and related materials.
3. Cloud server: which is deployed on the untrusted cloud, is responsible for storing archive files (in the form of ciphertext), archive description data (in the form of cipher-text) and archive feature data submitted by the local server, and is also responsible for executing archive search requests submitted by the local server.
4. Local server: which is deployed on the trusted local, responsible for strictly encrypting the archive files and archive description data submitted by an archive administrator, generating the corresponding archive feature data, and then submitting them to the cloud server for storage, and recording the corresponding encryption key data and setting parameters locally (i.e., responsible for running the confidentiality release model of archives on the cloud). In addition, it is also responsible for rewriting the archive search requests submitted by an archive inquirer, so that they can be correctly executed on the feature data of the cloud (to filter out most non-target records on the cloud) to ensure the accuracy and efficiency of archive search (i.e., responsible for running the confidentiality search model of archives on the cloud).

### 2.2. Design Goals

In order to satisfy the practical application requirements of confidentiality management of archives on the cloud, a confidentiality management model constructed based on the framework shown in Figure 1 should meet the following three constraints.

1.  Ensuring the security of archive data, which includes archive file security, archive data security and feature data security, i.e., from the encrypted archive files, encrypted archive data and feature data submitted by the local server, it is impossible for the cloud server to accurately know the original archive files and sensitive archive data.
2.  Ensuring the accuracy of archive search. With the help of archive feature data constructed by the confidential release model of archives on the cloud, each archive query operation defined on the archive data submitted by an archive inquirer can be effectively converted into a feature query operation defined on the feature data (i.e., Step 4 in Figure 1), so that the result returned by the cloud server by executing the feature query operation (i.e., the data returned by Step 5 in Figure 1) contains the real search result to ensure the accuracy of archive search.
3.  Ensuring the efficiency of archive search. With the help of feature data, the cloud server can eliminate most of the non-target records on the cloud by executing each feature query operation constructed by the confidential search model of archives on the cloud, so as to reduce the amount of archive data returned to the client (i.e., the data returned by Step 6 in Figure 1), and in turn, ensure the efficiency of archive data search.

## 3. Proposed Solution

### 3.1. Archive Confidentiality Model

On the basis of the framework of Figure 1, this paper constructs a confidentiality management model of digital archives on the cloud, which mainly includes two sub-models, i.e., a confidentiality release model of archives on the cloud, and a confidentiality search model of archives on the cloud. Here, the confidentiality release model corresponds to Steps 1 and 2 in Figure 1, i.e., the process of the local server to encrypt archive files and archive data released by an archive administrator, and attach archive feature data, which can be further shown in Figure 2. The description can be divided into the following four steps.

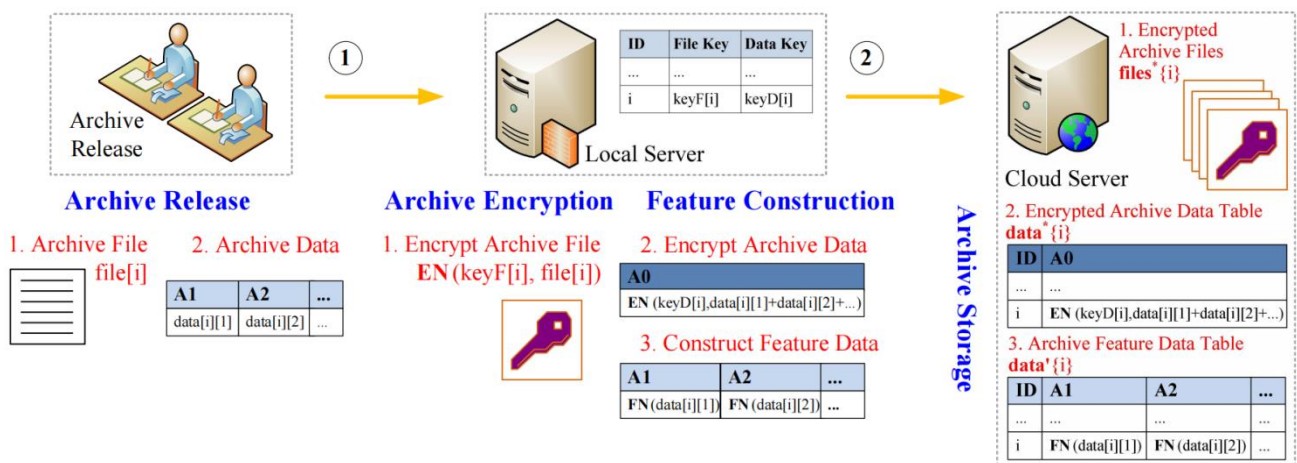

**Figure 2.** Implementation process of the confidentiality release model.

**Step 1.1.** Archive Release (executed by an archive administrator). An archive administrator releases an archive file and its corresponding archive description data through an archive management interface. The archive description data is denoted by $(\text{data}[i][1], \text{data}[i][2], \ldots)$, where $\text{data}[i][j]$ denotes a sensitive archive data item (i.e., some private data such as name,

ID number, phone number, home address, etc., which cannot be known to the cloud). The archive file is denoted by file[$i$] (usually which is an electronic scanning picture).

**Step 1.2.** Archive Encryption (executed by the local server). First, the local server generates an archive file key (denoted by keyF[$i$]) and an archive data key (denoted by keyD[$i$]) randomly. Then, using a traditional encryption algorithm (such as RSA, etc.), the local server strictly encrypts the archive file and archive description data submitted by an archive administrator, so as to obtain an encrypted archive file (denoted by file$^*$) and an encrypted archive data (denoted by data$^*$), which are, respectively, denoted by Equations (1) and (2).

$$\text{data}^*[i] = \mathbf{EN}\left(\text{keyD}[i], \text{data}[i][1] + \text{data}[i][2] + \ldots\right) \tag{1}$$

$$\text{file}^*[i] = \mathbf{EN}\left(\text{keyF}[i], \text{file}[i]\right) \tag{2}$$

Finally, the local server submits the encrypted archive file and the encrypted archive data to the cloud server for storage and stores the archive file key and archive data key locally (note that the secret keys are generated dynamically and randomly, and the secret keys of archive files are different from each other, and the keys of archive data are also different from each other).

**Step 1.3.** Feature Construction (executed by the local server). First, the local server generates the corresponding archive feature data (denoted by data$'$) for archive-sensitive data, which is denoted by Equation (3).

$$\left(\text{data}'[i][1] = \mathbf{FN}(\text{data}[i][1]), \text{data}'[i][2] = \mathbf{FN}(\text{data}[i][2]), \ldots\right) \tag{3}$$

Then, the local server submits the feature data to the cloud server for storage. The parameters related to feature construction are stored on the local server (note that the same archive data item uses the same feature parameter, and different items use different feature parameters).

**Step 1.4.** Archive Storage (executed by the cloud server). The cloud server stores the encrypted archive data and archive feature data in its archive database, as well as the encrypted archive files in its storage devices. Then, it establishes the associations (e.g., using URLs) between the archive data records of the database and the encrypted archive files.

The confidentiality release model of archives on the cloud corresponds to Steps 3 to 6 in Figure 1, i.e., the process of the local server to rewrite and replace each archive query operation defined on the archive description data released by an archive enquirer with a feature query operation defined on the corresponding archive feature data, and the process of decrypting and filtering the archive query result returned by the cloud server. The process can be further described in Figure 3, which can be divided into the following four steps.

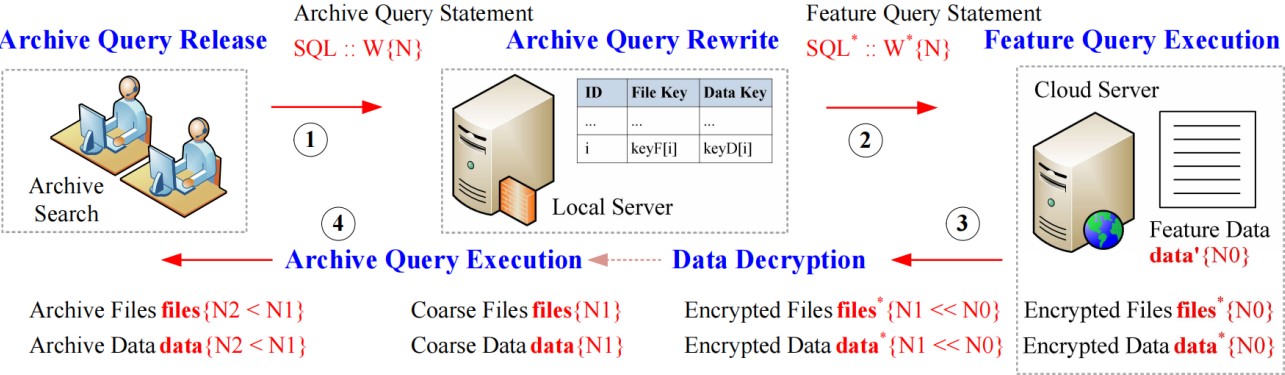

**Figure 3.** Implementation process of the confidentiality search model.

**Step 2.1.** Query Release (executed by an archive inquirer). An archive inquirer submits an archive query statement (defined on archive description data) through an archive query interface, to the local server. An archive query statement is mainly composed of a series of basic query conditions defined on archive data items and connected by logical operations. To this end, the basic query conditions of an archive query statement can be denoted by $(W[1], W[2], \ldots, W[N])$.

**Step 2.2.** Query Rewrite (executed by the local server). The local server converts each archive query statement defined on archive description data published by an inquirer into a feature query statement defined on the corresponding feature data and then submits it to the cloud server for execution. A feature query statement is mainly composed of a series of basic query conditions defined on feature data and connected by logical operations, which can be denoted by Equation (4).

$$(W^*[1] = TR(W[1]), W^*[2] = TR(W[2]), \ldots, W^*[N] = TR(W[N])) \tag{4}$$

**Step 2.3.** Query Execution (executed by the cloud server). The cloud server executes the feature query statement submitted by the local server on the feature dataset $data'\{N_0\}$ (where $N_0$ denotes the size of the feature dataset), and then returns the set of encrypted archive data $data^*\{N_1\}$ ($N_1 \ll N_0$) and the set of encrypted archive files $files^*\{N_1\}$ to the local server.

**Step 2.4.** Result Decryption (executed by the local server). For the encrypted archive dataset returned by the cloud server, in combination with the associated archive data keys saved by the local server, after decrypting the encrypted data, the local server obtains the corresponding plaintext archive dataset denoted by $data\{N_1\}$. Second, the local server executes the original archive query statement issued by the archive manager on the plaintext archive dataset to obtain the target archive dataset denoted by $data\{N_2\}$ ($N_2 < N_1$). Let $files^*\{N_2\}$ denote the set of the encrypted archive files associated with the dataset $data\{N_2\}$. Finally, the local server decrypts the ciphertext file set in combination with the locally stored file keys to obtain the corresponding plaintext archive file set $files\{N_2\}$, and return the archive file set $files\{N_2\}$ and the archive data set $data\{N_2\}$ to the client.

*3.2. Feature Construction and Query Rewriting*

From the previous section, we can see that feature construction (Step 1.3) is the key to the confidentiality release model of archives on the cloud, and query rewriting (Step 2.3) is the key to the confidentiality search model of archives on the cloud. Moreover, we can see the query rewriting strategy is closely dependent on the feature construction strategy. To this end, this section first gives a simple and effective strategy for feature construction and then constructs a corresponding strategy for query rewriting accordingly. Note that the types of archive data items are mainly divided into numerical type (e.g., real number, date, etc.) and text type (i.e., character string). The strategy of feature construction proposed in this paper can be applied to the two data types at the same time, whose process mainly includes the following three steps.

**Step 3.1.** Suppose that an archive-sensitive data item $\mathbf{A}_0$ contains $n$ basic units, respectively, denoted by $\mathbf{A}_1, \mathbf{A}_2, \ldots, \mathbf{A}_n$. For each basic unit $\mathbf{A}_k$, we divide the domain composed of all its possible values into $N_k$ subdomains, denoted by $\mathbb{D}_1^k, \mathbb{D}_2^k, \ldots, \mathbb{D}_{N_k}^k$, which should meet the constraints.

1. None of the subdomains is an empty set, i.e., $\mathbb{D}_i^k \neq \oslash$;
2. Any two subdomains do not overlap, i.e., $\mathbb{D}_i^k \cap \mathbb{D}_j^k = \oslash$;
3. The union of all subdomains is equal to the domain itself of the basic unit, i.e., $\cup \mathbb{D}_i^k = \mathbb{D}_k$.

**Step 3.2** Assign an identifier to each subdomain $\mathbb{D}_i^k$, denoted by $\varpi_k\left(\mathbb{D}_i^k\right)$. All the identifiers should meet the following constraints.

1. Each identifier itself is selected from the domain $\mathbb{D}_k$ of the basic unit, i.e., $\varpi_k\left(\mathbb{D}_i^k\right) \in \mathbb{D}_k$;

2. All identifiers remain in order, i.e., if $\varpi_k\left(\mathbb{D}_i^k\right) \geq \varpi_k\left(\mathbb{D}_j^k\right)$, then $\forall a \in \mathbb{D}_i^k \; \forall b \in \mathbb{D}_j^k \to a \geq b$;

3. The length of each identifier is equal to that of the maximum value of the domain $\mathbb{D}_k$, i.e., $\left|\varpi_k\left(\mathbb{D}_i^k\right)\right| = |\mathbf{max}(\mathbb{D}_k)|$.

Based on the settings of the above two steps, any specific value $a_k$ of the subdomain $\mathbf{A}_k$ can be mapped to an identifier value of the same length with $a_k$, i.e., a feature mapping function is determined, denoted by $\mathbf{FN}_k(a_k) = \varpi_k\left(\mathbb{D}_i^k\right)$, where $\mathbb{D}_i^k$ is the subdomain which contains $a_k$. Now, based on the settings of the above two steps, we have actually determined $n$ feature mapping functions for the archive-sensitive data item $\mathbf{A}_0$, which are denoted by $\mathbf{FN}_1, \mathbf{FN}_2, \ldots, \mathbf{FN}_n$ (corresponding to the basic units $\mathbf{A}_1, \mathbf{A}_2, \ldots, \mathbf{A}_n$, respectively).

**Step 3.3** For any value $a$ of the archive data item $\mathbf{A}_0$, based on the settings of the above two steps, we assume that the values corresponding to the basic units of $\mathbf{A}_0$ are $a_1, a_2, \ldots, a_n$, respectively, i.e., $a = a_1 a_2 \ldots a_n$. Then, based on the functions $\mathbf{FN}_1, \mathbf{FN}_2, \ldots, \mathbf{FN}_n$, it can be mapped to a new feature value (i.e., feature data), denoted by Equation (5).

$$a' = \mathbf{FN}(a) = \mathbf{FN}_1(a_1)\, \mathbf{FN}_2(a_2) \ldots \mathbf{FN}_n(a_n) \tag{5}$$

**Example 1.** *Take the archive-sensitive data item Name as an example to briefly describe the construction process of feature data. Here, we assume that the maximum length of the name field is 8 Chinese characters (i.e., it contains 8 basic units). First, let us consider the first basic unit. Note that there are 20902 common Chinese characters, and their UNICODE codes are between 0X4E00 and 0X9FA5. To this end, we simply divide the value range of Chinese characters into 209 subdomains (so the size of each subdomain is equal to 100) by using an equal-width strategy, respectively, denoted by $\mathbb{D}_1^1, \mathbb{D}_2^1, \ldots, \mathbb{D}_{209}^1$ (Step 3.1). Then, we assign an identifier for each subdomain according to the following strategy $\varpi_1\left(\mathbb{D}_k^1\right) = k$ (Step 3.2),*

$$\varpi_1\left(\mathbb{D}_1^1\right) = 0X0001; \; \varpi_1\left(\mathbb{D}_2^1\right) = 0X0002; \ldots; \; \varpi_1\left(\mathbb{D}_{209}^1\right) = 0X00D1 \tag{6}$$

To simplify the presentation, for the remaining seven basic units of the data item, we apply the same subdomain division and identifier assignment strategies as the first unit, i.e., $\varpi_1 = \varpi_2 = \ldots = \varpi_8$, so we have that $\mathbf{FN}_1 = \mathbf{FN}_2 = \ldots = \mathbf{FN}_8$. Now, for any given specific name, we can generate its corresponding feature data value. For example, for a Chinese name whose UNICODE encode is 0X8BF8 0X845B 0X4EAE, its feature data value after feature construction is as 0X009A 0X008B 0X0002.

Based on the settings of Steps 3.1 and 3.2, we can see that feature data has the same length and format as its corresponding archive data, so feature data generated in Step 3.3 can be directly stored in the field $\mathbf{A}_0$ of archive data tables. So far, after feature mapping, feature data (instead of archive data) is stored in archive-sensitive data item fields of the cloud database. However, this makes the query operations defined on the archive data items issued by an archive inquirer no longer correctly executed in the cloud database. To this end, the purpose of query rewriting is to transform each archive query condition into a feature query condition defined on feature data. Since an archive query statement is mainly composed of a series of basic query condition items connected by logical operators, below, we briefly discuss how to rewrite three kinds of basic archive query condition items (i.e., equivalent query, implication query and range query), and then introduce Algorithm 1 to show how an archive query statement is rewritten.

---

**Algorithm 1** Query Rewriting.

---

**(1) Input:** an archive query statement; **(2) Output:** a feature query statement
01. Divide the archive query statement into a series of basic archive query condition items;
02. **FOREACH** basic archive query condition item **DO**
03.   **IF** the item is an equivalent condition item **THEN**
04.     **CALL** Conversation 1.1 to convert it into a feature equivalent query condition;
05.   **ELSEIF** the item is an implication condition item **THEN**
06.     **CALL** Conversation 1.2 to convert it into a feature implication query condition;
07.   **ELSEIF** the item is a range condition item **THEN**
08.     **CALL** Conversation 1.3 to convert it into a feature range query condition;
09.   **END IF**
10. **END FOR**
11. **RETURN** a feature query statement constructed based on the feature query conditions.

---

**Conversion 1.1.** Equivalent Query Conversion: A basic equivalent query conditional item defined on an archive-sensitive data item $\mathbf{A}_0$ can be generally expressed as $\mathbf{A}_0 = a_0$, where $a_0 = a_1 a_2 \ldots a_n$ represents a constant defined on the archive data item $\mathbf{A}_0$. Then, the archive equivalent query condition item can be converted into a feature equivalent query condition, denoted by Equation (7)

$$\mathbf{TR}(\mathbf{A}_0 = a_0) \Rightarrow \mathbf{A}_0 = \mathbf{FN}(a_0) \Rightarrow \mathbf{A}_0 \\ = \mathbf{FN}_1(a_1)\, \mathbf{FN}_2(a_2) \ldots \mathbf{FN}_n(a_n) \tag{7}$$

An implication query conditional item is generally constructed based on the predicate LIKE, whose general syntax can be represented by LIKE<match string>. The matching string can contain a variety of wildcards, among which % (which denotes to match a character string of any length) is the most representative wildcard. Below, we only discuss the conversion of a left direction implication query condition based on the wildcard %.

**Conversion 1.2.** Implication Query Conversion: A basic implication query condition item defined on an archive-sensitive data item $\mathbf{A}_0$ can be generally expressed as $\mathbf{A}_0$ **LIKE** $a_0$%. Based on the setting of Step 3.1, we assume that $a_0$ completely covers $k$ basic units (i.e., $\mathbf{A}_1, \mathbf{A}_2, \ldots, \mathbf{A}_k$) of the data item $\mathbf{A}_0$, and the values corresponding to the $k$ basic units are $a_1, a_2, \ldots, a_k$, respectively. Then, the implication query condition item can be converted into a feature implication query condition, denoted by Equation (8).

$$\mathbf{TR}(\mathbf{A}_0 \ \mathbf{LIKE} \ a_0\%) \Rightarrow \mathbf{A}_0 \ \mathbf{LIKE} \ \mathbf{FN}_1(a_1)\, \mathbf{FN}_2(a_2) \ldots \mathbf{FN}_k(a_k)\% \tag{8}$$

**Conversion 1.3.** Range Query Conversion: A range query condition item defined on an archive-sensitive data item $\mathbf{A}_0$ can be generally expressed as $\mathbf{A}_0 \geq a_0$. Then, the range query condition item can be converted into a feature range query condition, denoted by (9).

$$\mathbf{TR}(\mathbf{A}_0 \geq a_0) \Rightarrow \mathbf{A}_0 \geq \mathbf{FN}(a_0) \Rightarrow \mathbf{A}_0 \geq \mathbf{FN}_1(a_1)\, \mathbf{FN}_2(\mathbf{a}_2) \ldots \mathbf{FN}_n(a_n) \tag{9}$$

**Example 2.** *Take querying the archive-sensitive data item Name as an example to briefly describe the query rewriting process. Assume that an archive inquirer wants to query the digital archive information from the persons named "ZhangSan" or surnamed "Liu". Then, an archive query statement defined on archive data submitted from a query interface can be presented as follows*
  **SELECT \* FROM** DATA **WHERE** Name = "ZhangSan" **OR** Name **LIKE** "Liu%"
  *It can be seen that the statement contains two basic archive query conditional items. Then, the feature query statement generated by the local server after equivalent query transformation and implication query transformation can be presented as follows*
  **SELECT \* FROM** DATA **WHERE** Name = **TR**("ZhangSan") **OR** Name **LIKE** TR("Liu")%

From Examples 1 and 2, we can see that the query rewriting strategy is closely dependent on the feature construction strategy, but the converted feature query statement can be directly executed by the cloud database, and most of the non-target records can be filtered

out on the cloud accordingly, thereby ensuring the accuracy and efficiency of archive search (please refer to the accuracy analysis and efficiency analysis in Section 4 for detail).

## 4. Analysis and Evaluation

### 4.1. Security Analysis

Generally, the cloud server is considered to be honest but curious, i.e., although it can follow the protocol specifications related to cloud services, it remains curious about archive files and archive data. In other words, the cloud server is not trusted. To this end, in this section, we theoretically analyze the security of the confidentiality management model of digital archives on the cloud, including archive file security, archive data security, and feature data security, i.e., analyze the possibility that the untrusted cloud server obtains sensitive archive information, according to the encrypted archive files, encrypted archive data and archive feature data submitted by the local server.

**Observation 1.1.** The confidentiality management model proposed in this paper can effectively ensure the security of digital archives on the cloud. The model adopts a traditional encryption algorithm to strictly encrypt the digital archive files (in the form of images), and the keys are stored in a trusted local server. As a result, the cloud server can obtain neither the keys nor the archive content based on the encrypted archive files.

**Observation 1.2.** The confidentiality management model proposed in this paper can effectively ensure the security of sensitive data of digital archives on the cloud. The model adopts a traditional encryption algorithm to strictly encrypt archive-sensitive data, and the keys are stored in the trusted local server. As a result, the cloud server can obtain neither the keys, nor the archive-sensitive data based on the encrypted archive data.

**Explanation**: Observations 1.1 and 1.2 are easy to be explained. The security of traditional encryption algorithms has been proved by a lot of practice, i.e., without knowing the secret key, it is almost impossible for an attacker to directly obtain the plaintext corresponding to the ciphertext. However, the secret key is stored in the trusted local server, which cannot be obtained by the cloud server.

In order to support archive search, the confidentiality management model proposed in this paper introduces feature data for archive data, which inevitably reflects some key characteristics (such as comparability and similarity) of archive data, consequently, leading to some risk of privacy leakage. This risk can be measured by the possibility of an attacker successfully guessing the corresponding archive data based on the feature data.

**Observation 1.3.** The confidentiality management model proposed in this paper can effectively ensure the security of archive feature data on the cloud. Below, we analyze the probability of the cloud successfully guessing the archival data based on feature data under the worst case. At this time, we assume that the attacker has completely understood the feature construction process of an archive data item and obtained the relevant feature parameters on the local server, i.e., the attacker has mastered the feature function **FN**. Assume that the archive data item contains $n$ basic units and the value range $\mathbb{D}_k$ of each basic unit is divided into $N_k$ subdomains. Now, given any feature data $a'$, the possibility of the attacker successfully guessing the corresponding plaintext data $a$ can be measured as the Equation (10).

$$\mathbf{PR}\left(a \mid a'\right) = \frac{\text{the size of the domain of } a'}{\text{the size of the domain of } a} = \frac{(N_1 \cdot N_2 \cdot \ldots \cdot N_n)}{|\mathbb{D}_1| \cdot |\mathbb{D}_2| \cdot \ldots \cdot |\mathbb{D}_n|} = \frac{N}{|\mathbb{D}|} \qquad (10)$$

It can be seen that the range of the value of $N$ (equal to the accumulation of the numbers of the subdomains of all the basic units) is $[1, |\mathbb{D}|]$, and the feature data security can be controlled by adjusting the value of $N$. Moreover, it can be seen that when the value of $N$ is smaller (i.e., when each basic unit is roughly divided), the possibility of the attacker obtaining the plaintext would be very small, i.e., even if the cloud server has obtained the feature mapping function, it is difficult to further obtain the archive data according to feature data. Below, the value of $N/|\mathbb{D}|$ is referred to as feature threshold. The larger the feature threshold, the worse the security of feature data, and the smaller the feature

threshold, the better the security of feature data. Moreover, the feature threshold value would affect the efficiency of archive search (see Section 4.3 for detail). Based on the above three observations, it can be further concluded that the confidentiality management model of archives on the cloud constructed in this paper can effectively ensure the security of archive files, archive data and feature data, i.e., it has good security.

*4.2. Accuracy Analysis*

In this section, we analyze the accuracy of the archive confidentiality search model proposed in this paper. In the mode, with the help of feature data, each query operation defined on archive data would be transformed into a feature query operation defined on feature data. In order to ensure the accuracy of archive search, the result returned from the cloud by executing each feature query operation has to contain the exact result corresponding to the archive query operation. To this end, below, we first introduce Observation 2.1 and Observation 2.2 to demonstrate that each feature query condition obtained based on Conversations 1.1 to 1.3 can ensure the accuracy of archive search.

**Observation 2.1.** Let W denote an implication query condition before conversion, and $W^*$ the feature implication query condition after conversion (Conversion 1.2). Then, for any archive data $a_1 a_2 \ldots a_n$, if its corresponding feature data $a'_1 a'_2 \ldots a'_n$ $(a'_k = \mathbf{FN}_k(a_k))$ meets the condition $W^*$, it certainly meets the condition W.

**Explanation**: An implication query condition is only targeted for textual data (not for numeric data). Let $b_1 b_2 \ldots b_m$ denote the text constant associated with the implication query condition W, and $b'_1 b'_2 \ldots b'_m$ denote its feature data. Because the feature data $a'_1 a'_2 \ldots a'_n$ meets the feature implication query condition $W^*$ (i.e., it contains $b'_1 b'_2 \ldots b'_m$), i.e., it exists $k \leq n$, such that $a'_1 a'_2 \ldots a'_k = b'_1 b'_2 \ldots b'_m$. Based on Conversation 1.2, we can conclude that the text constant $b_1 b_2 \ldots b_m$ corresponding to the feature data $b'_1 b'_2 \ldots b'_m$ is certainly contained in the archive data $a_1 a_2 \ldots a_n$ corresponding to the feature data $a'_1 a'_2 \ldots a'_n$, i.e., the archive data $a_1 a_2 \ldots a_n$ meets the implication query condition W.

**Observation 2.2.** Let W denote a range query condition before conversion, and $W^*$ the feature range query condition after conversion (Conversion 1.3). Then, for any archive data $a_1 a_2 \ldots a_n$, if its corresponding feature data $a'_1 a'_2 \ldots a'_n$ meets the condition $W^*$, it certainly meets the condition W.

**Explanation**: A range condition is targeted for both textual data and numeric data. Let $b_0$ denote the text constant associated with the range query condition W. For any given archive data $a_0$, it may not be consistent with the length of the constant $b_0$. In this situation, it can be right-padded with zeros (encode values) for short text data, or left-padded with zeros (integer values) for numeric data, to make both with the same length. Let $a_0 = a_1 a_2 \ldots a_n$ and $b_0 = b_1 b_2 \ldots b_n$, and $a'_0 = a'_1 a'_2 \ldots a'_n$ and $b'_0 = b'_1 b'_2 \ldots b'_n$ denote the feature data corresponding to $a_0$ and $b_0$. Because the feature data $a'_1 a'_2 \ldots a'_n$ meets the range condition $W^*$, i.e., $a'_1 a'_2 \ldots a'_n \geq b'_1 b'_2 \ldots b'_n$ (it is assumed to be greater than the comparison), we conclude that there certainly exists that $1 \leq k \leq n$ such that $a'_1 = b'_1, a'_2 = b'_2, \ldots, a'_k \geq b'_k$. Based on the constraints of the previous feature construction strategy (Step 3.2), we can further conclude that $a_1 = b_1$, $a_2 = b_2$, $\ldots, a_k \geq b_k$ ( $a_1 a_2 \ldots a_n \geq b_1 b_2 \ldots b_n$), i.e., the archive data meets the range query condition W.

An equivalence query can be regarded as a special implication query or a special range query, so its accuracy analysis is no longer presented. Note that equivalence query, implication query and range query are three kinds of the most common basic conditions, and other query conditions can be completed directly or indirectly by means of them. Therefore, based on the above observations, we can further conclude that various query operations defined on archive-sensitive data can be converted into feature query operations defined on feature data, and the results returned from the cloud by executing these feature query operations certainly contain the real results corresponding to the original query operations, i.e., the confidentiality management model of archives on the cloud proposed in this paper can effectively ensure the accuracy of archive search.

*4.3. Efficiency Evaluation*

In this section, we evaluate the efficiency of the archive confidentiality search model through experiments, i.e., whether the feature query can filter out most of the non-target data on the cloud, so as to improve the archive search efficiency. The experiments were run on a randomly generated table of one million digital archive data records. The experiment selects two sensitive archive data items (name and birthday), which are text type and numerical type, respectively. From the archival data (i.e., Name and Birthday), which are text type and numerical type, respectively. From the search process of archive data shown in Figure 3, it can be seen that the search efficiency of feature data depends on the filtering effect of the feature query operations obtained by query transformation on the non-target records on the cloud. To this end, we introduce the following definition to measure search efficiency.

**Definition 1.** *Let W denote a query condition before transformation, and $W^*$ denote the feature query condition defined on feature data after transformation. Let $N_0$ denote the number of archive records, $N_2$ denote the number of records that meet the archive query W, and $N_1$ denote the number of records that meet the feature query $W^*$. Then, the search efficiency of feature data can be measured by the filtering effect of the feature query on the non-target records, i.e., $\textbf{FR}\ (W^*,\ W) = (N_0 - N_1)/(N_0 - N_2)$.*

The efficiency evaluation is divided into three groups of experiments, i.e., range query on numeric data, range query on textual data, and implication query on textual data. (1) The first group of experiments aims to evaluate the efficiency of range query operations on numeric data. The experimental results are shown in Figure 4, where the abscissa represents the feature threshold (see Observation 1.3 for detail), and the ordinate represents the query efficiency. It can be seen that the filtering effect of feature range query operations on the non-target records would become worse as the feature threshold decreases. This is because the decrease in the feature threshold would increase the number of possible plaintexts corresponding to each feature data value, resulting in a decrease in the query efficiency measure. However, even if the feature threshold is smaller (e.g., less than $2^{-12}$), each feature range query operation can still filter out most of the non-target records (greater than 0.99), thereby, reducing the scale of the records returned to the client, and in turn, greatly improving the range query efficiency. (2) The second group of experiments aims to evaluate the efficiency of range query operations on textual data, and the experimental results are shown in Figure 5. It can be seen that the change trend of the range query efficiency measure of textual data with respect to the feature threshold is consistent with that of numerical data. (3) The third group of experiments aims to evaluate the implication query efficiency of textual data, and the experimental results are shown in Figure 6. It can be seen that with the decrease in the feature threshold, the filtering effect of feature implication query operations on the non-target records would become worse (the change trend is basically the same as that of textual range query operations); however, compared with textual range query operations, implication feature query operations have a better filtering effect on non-target records (i.e., having greater values for the efficiency measure). This is because the target record set of an implication query is extremely smaller (usually thousands), while the target record set of a range query is extremely larger (usually hundreds of thousands).

From the three groups of experiments mentioned above, we can draw a conclusion that both for implication query conditions or range query conditions, both for textual data and numerical data, by executing the feature query conditions obtained through feature transformation, the cloud can filter out most non-target records (greater than 0.99), thereby reducing the scale of records returned to a client, and in turn, effectively reducing the time overhead of archive search, i.e., feature data has good search efficiency.

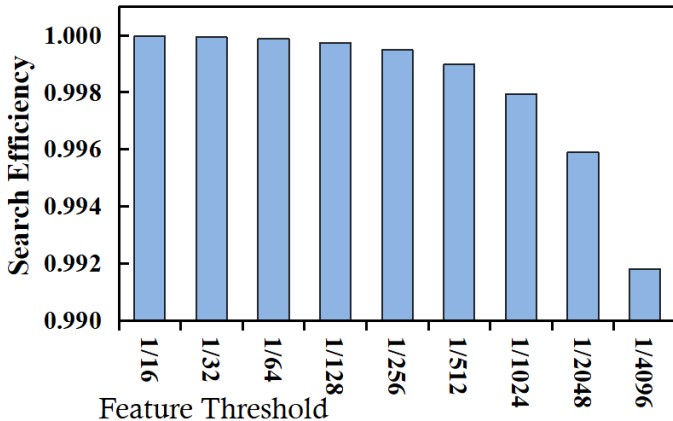

**Figure 4.** Evaluation results for numeric data range query efficiency.

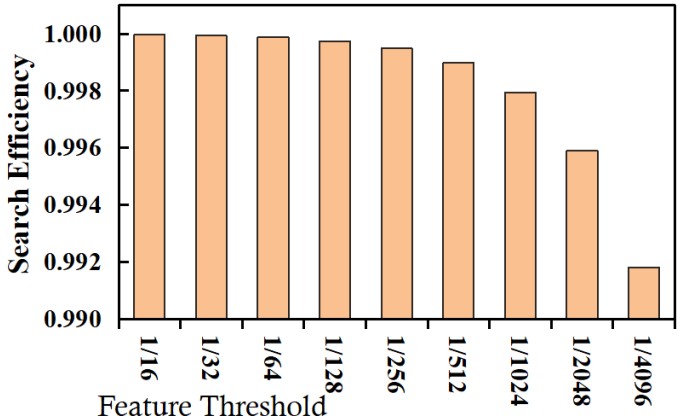

**Figure 5.** Evaluation results for textual data range query efficiency.

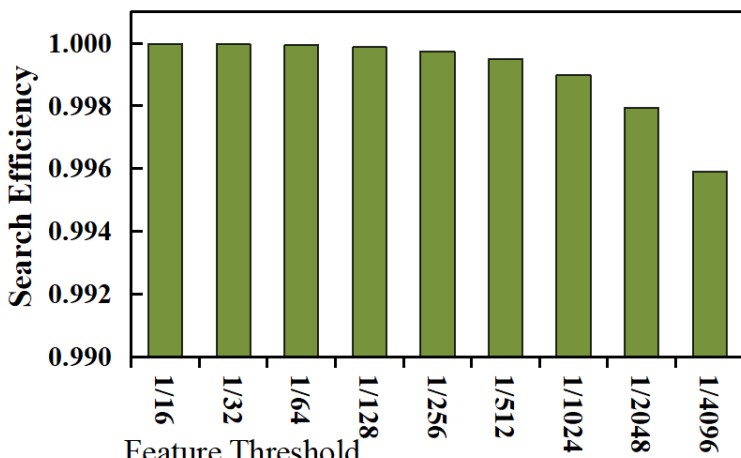

**Figure 6.** Evaluation result for textual data implication query efficiency.

Finally, Table 1 presents a brief comparison between our proposed solution and other related ones mentioned in Section 1.1. From the table, we see that compared with others, our solution has better overall performance in terms of confidentiality, accuracy, efficiency and availability, which demonstrates again that our solution can well meet the goals presented in Section 2.2. At last, it should be pointed out that although the solution of this paper is targeted at the confidentiality management of digital archives in a cloud

environment, it can be transferred to other problems of data confidentiality management as well, such as multimedia data management [54,55], knowledge management [56–60], and series management [61–64].

**Table 1.** A comparison between our solution and other related ones.

| Methods | Confidentiality | Accuracy | Efficiency | Availability |
|---|---|---|---|---|
| Our solution | Good | Good | Good | Good |
| Identity Authentication | Not good | Good | Good | Good |
| Access Control | Not good | Good | Good | Good |
| Encryption | Good | Good | Good | Not good |

## 5. Conclusions

Aiming at the problem of confidentiality management of digital archives in a cloud environment, this paper constructs an archive release model and an archive search model, whose basic idea is to strictly encrypt all archive files and their corresponding archive data on a trusted local server, before they are submitted to the cloud for storage, to ensure the security of archive data on the untrusted cloud. In order to solve the problem of archive search, the solution also adds additional feature data to the encrypted archive data, so that each query operation defined on archive data can be executed on the cloud, thereby, greatly improving the efficiency of archive data query, and in turn ensuring the effectiveness of archive search. This paper presents a valuable research attempt on the problem of confidentiality management of archives on the cloud. The solution proposed in this paper can effectively balance the security of archive data and the effectiveness of archive search, i.e., it can ensure the security of sensitive archive information on the untrusted cloud, without affecting the efficiency and accuracy of archive search. It has positive significance for promoting the further application and development of cloud computing technology in archives management.

However, the proposal of this paper is not the end of our work. In future work, we will try to further study some problems, e.g., (1) how to simplify the archive release model and the archive search model to reduce the workload of the local server; (2) how to design different feature construction schemes for different archive data types, to improve the efficiency and security; and (3) the practical implementation of the proposed method in a management system of digital archives in a cloud environment.

Finally, Table 2 describes some key symbols used in the paper.

**Table 2.** Symbols and their meanings.

| Symbols | Meanings |
|---|---|
| $\text{data}[i]$ | A sensitive archive data record |
| $\text{data}^*[i]$ | An encrypted archive data record |
| $\text{data}'[i]$ | An archive feature data record |
| $W[i]$ | A basic query condition defined on archive data |
| $W^*[i]$ | A basic query condition defined on feature data |
| $\mathbf{A}_k$ | A basic unit of an archive-sensitive data item |
| $\mathbb{D}_i^k$ | A subdomain of the domain of the basic unit $\mathbf{A}_k$ |
| $\varpi_k\left(\mathbb{D}_i^k\right)$ | An identifier of the subdomain $\mathbb{D}_i^k$ |
| $\mathbf{FN}_k$ | A feature mapping function for the basic unit $\mathbf{A}_k$ |
| $a$ | A value of an archive data item |
| $\mathbf{FN}(a)$ | A feature mapping function for an archive data item |
| $a'$ | A feature value of an archive data item |
| TR | A condition conversion function |

**Author Contributions:** Methodology, S.X.; writing—original draft preparation, J.X. software, W.Y.; writing—review and editing, Z.W.; software, H.C. All authors have read and agreed to the published version of the manuscript.

**Funding:** The work is supported by the key project of Humanities and Social Sciences in Colleges and Universities of Zhejiang Province (No 2021GH017), Humanities and Social Sciences Project of the Ministry of Education of China (No 21YJA870011), Zhejiang Philosophy and Social Science Planning Project (No 22ZJQN45YB) and National Social Science Foundation of China (No 21FTQB019).

**Institutional Review Board Statement:** Not applicable.

**Data Availability Statement:** Not applicable.

**Conflicts of Interest:** The authors declare no conflict of interest.

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
