# Peer review of "An Effective Model of Confidentiality Management of Digital Archives in a Cloud Environment"

_electronics, doi:10.3390/electronics11182831_

Round 1
Reviewer 1 Report
Here is the list of minor remarks:1. Please provided sources under the labels of the figures.
2. Provide information about equation number at the end of the line, e.g. "... which are respectively denoted by (1):
...
3. Step 2.2) Equation without (3) - there are more below, please correct it.
4. Please change "5. Conclusion" to "5. Conclusions".
5. Please elaborate the point 5: provide information about further works.
6. Check references: some of them are not ending with the dot.
Author Response
Response to Reviewer 1 Comments
Point 1: Please provided sources under the labels of the figures.
Response 1: All the figures (Figures 1 to 4) have been replaced by clearer sources.
Point 2: Provide information about equation number at the end of the line, e.g. “... which are respectively denoted by (1)”.
Response 2: The problem has been revised. For each the equation in the manuscript, the equation number and its related information have been provided.
Point 3: Step 2.2) Equation without (3) - there are more below, please correct it..
Response 3: The problem has been revised. For each the equation in the manuscript, the equation number have been provided.
Point 4: Please change "5. Conclusion" to "5. Conclusions".
Response 4: The problem has been revised.
Point 5: Please elaborate the point 5: provide information about further works.
Response 5: The problem has been revised. In the new manuscript, we have rewritten the Conclusion section. At the end of the Conclusion section, we have added a new paragraph to discuss the future work of this paper.
Point 6: Check references: some of them are not ending with the dot.
Response 6: The problem has been revised. In the new manuscript, we have added some new references.

Reviewer 2 Report
The authors present work on how to deal with the confidentiality of digital archives in the cloud.
The article is well structured and well written.
Author Response
Response to Reviewer 2 Comments
Point 1: The authors present work on how to deal with the confidentiality of digital archives in the cloud. The article is well structured and well written.
Response 1: Thanks.
Reviewer 3 Report
Overall, the authors have made a good attempt. However, the proposed method does not adequately describe their data. The results are not supported by any theoretical/mathematical reasons. Readers will fail to understand the scientific contribution of this research. The authors should justify the effectiveness of the proposed technique theoretically.
1. The abstract does not discuss the finding of the research, this paper looks like a survey paper. The author should include the research findings in this section.
2. Abbreviations and nomenclature should be maintained for better understanding.
3. Related work section is precious but the presentation (taxonomy) must be improved.
4. The author must differentiate data* and data#.
5. Equation 2 is incomplete in RHS.
6. After equation 2, we couldn't find any equation number. The author should work on this.
7. Callout and citations are random in many places, correct it.
8. The author used many variables without any purpose, limit the usage of variable and provide the flow for reader understanding.
9. Section 3.2: This portion is a bit confusing, the author must include pseudocode for better understanding.
10. The author used a few local languages inside the example. The author should provide the meaning of those words.
11. Analysis section is weak, the observation and analysis look like a survey. The author should improve the analysis by adding state of the art discussion.
Author Response
Response to Reviewer 3 Comments
Point 1: The abstract does not discuss the finding of the research, this paper looks like a survey paper. The author should include the research findings in this section.
Response 1: The problem has been revised. In the new manuscript, we have rewritten the abstract section to add some discussion on the finding of the research.
Point 2: Abbreviations and nomenclature should be maintained for better understanding.
Response 2: The problem has been revised. For better understanding, we have revised the abbreviations and presentations carefully in our new manuscript.
Point 3: Related work section is precious but the presentation (taxonomy) must be improved.
Response 3: The problem has been revised. We have rewritten the related work section (Section 1.1) to improve the presentation, and added some new references.
Point 4: The author must differentiate data* and data#.
Response 4: The problem has been revised. We have added some description on the two symbols, where data* denotes encrypted archive data, and data’ denotes its feature data.
Point 5: Equation 2 is incomplete in RHS.
Response 5: The problem has been revised.
Point 6: After equation 2, we couldn't find any equation number. The author should work on this.
Response 6: The problem has been revised. In the new manuscript, we have added the number for each equation.
Point 7: Callout and citations are random in many places, correct it.
Response 7: The problem has been revised. In the new manuscript, all the citations are well-ordered.
Point 8: The author used many variables without any purpose, limit the usage of variable and provide the flow for reader understanding.
Response 8: The problem has been revised. In the new manuscript, the usage of variables has been limited.
Point 9: Section 3.2: This portion is a bit confusing, the author must include pseudocode for better understanding.
Response 9: The problem has been revised. In the new manuscript, we have added pseudocode (Algorithm 3.1) for better understanding Section 3.2.
Point 10: The author used a few local languages inside the example. The author should provide the meaning of those words.
Response 10: The problem has been revised. In the new manuscript, we have inserted some footnotes to explain the meaning of the words from local langauge.
Point 11: Analysis section is weak, the observation and analysis look like a survey. The author should improve the analysis by adding state of the art discussion.
Response 11: The problem has been revised. In the revised Section 4 of the new manuscript (see the end of Section 4), we have added some analysis by introducing a brief comparison between our proposed solution and other related ones mentioned in Section 1.1.

Round 2
Reviewer 3 Report
The author has revised the article substantially but the following comments should be addressed too:
1. Findings of the reach should be included in the abstract section.
2. Equation 6-9 call out is missing.
3. Nomenclature and abbreviation tables are required to be included at the end of the paper.
Author Response
Response to Reviewer 3 Comments
Point 1: Findings of the reach should be included in the abstract section.
Response 1: The findings of the reach haven been included in the abstract section, i.e., “…, Finally, both theoretical analysis and experimental evaluation demonstrate the good performance of the proposed solution. The result shows that compared with others, our solution has better overall performance in terms of confidentiality, accuracy, efficiency and availability, which can improve the security of archive sensitive data on the untrusted cloud without compromising the performance of an existing archive management system.”
Point 2: Equation 6-9 call out is missing.
Response 2: The problem has been revised. The number 6-9 has been cited.
Point 3: Nomenclature and abbreviation tables are required to be included at the end of the paper.
Response 3: The problem has been revised. At the end of the paper, Table 2 has been added to describe some key symbols used in this paper.